# OpenReview forum: "LLM Lies: Hallucinations are not Bugs, but Features as  Adversarial Examples"
_ICLR.cc/2024/Conference — ICLR 2024 Conference Withdrawn Submission_

### Official Review · Reviewer_RvtD · 2023-10-27

**Soundness:** 1 poor
**Presentation:** 4 excellent
**Contribution:** 2 fair
**Rating:** 3
**Confidence:** 4

**Summary:**

The paper adapts gradient-based attacks known from discriminative image and text models to large language models. It can modify tokens in the input so that they produce wrong answers of the adversary’s choosing to factual questions.

**Strengths:**

It is of interest to the community that gradient-based adversaries can induce hallucinations and this paper develops an interesting method to do so.

**Weaknesses:**

It is unclear if the attacks transfer between models when they are generated on one model. For example, do attacks on Llama Chat work against ChatGPT?

The idea that gradient based adversaries can alter the output of a language model has been explored in several other works, for example: https://arxiv.org/abs/2307.15043

The paper severely overclaims in its title and its intro. They offer no discussion, empirical explorations or anything to back up their claim that hallucinations are fundamental features of language models.

**Questions:**

In the objective of Equation (4), do the authors do a forward pass that infers all tokens of the target answer at once? More generally, how do they compute a gradient if the LLM is predicting one token at a time?

What do their authors base their claim that hallucinations are fundamental to LLMs on?

---

### Official Review · Reviewer_NSE9 · 2023-11-01

**Soundness:** 2 fair
**Presentation:** 3 good
**Contribution:** 2 fair
**Rating:** 3
**Confidence:** 3

**Summary:**

In this paper, the author propose a gradient-based adversarial attack (test-time) that adversarially shifts the predicted tokens to a target hallucinated response given out-of-distribution or perturbed input prompts. Also, the author explores the features of the attacked adversarial prompts and propose an entropy-based thresholding scheme to reject the hallucinated prompts.

**Strengths:**

1. The paper has clear definition of concepts, and easy-to-understand formulation/algorithm for their proposed adversarial attacks.
2. The paper provides very intuition graphs and diagrams with examples that demonstrates the proposed attacks. In particular, the provided sample prompts are intuitive to understand.
3. The paper is generally well-written.

**Weaknesses:**

1. I find the proposed scenario of hallucination, i.e. adversarial prompts -> hallucinated response to be ill-motivated. From an end-user perspective, the inputs under either the weak semantic attack or OoD attack are non-sensical. The user of the LLM should simply write a legit sentence if they noticed what they wrote.
- For example, one improvement that can be made is to use some NLP based metrics instead of p-norm.
- The paper might want to look into this review paper [1] to formulate more realistic problem setup.
2. From my understanding, the most common scenario for hallucination is for unperturbed prompts to have a hallucinated response. The paper is probably more relevant to hallucination generation than hallucination itself.
3. "Hallucinations are not bugs, but features" is a very ambitious statement that the paper cannot fully support. I believe this statement is partially inspired from [2]. However, some logics are missing.
- To identify hallucination as a fundamental property of the LLM model, the author should empirically (ideally theoretically) show that why hallucination is unavoidable. To mitigate it, what is the tradeoff. For example, we might have to sacrifice some performance.
- For example, the paper might want to adopt a similar strategy to [2] by defining what are robust, non-robust features, and show that the LLM performance rely on both features for high accuracy.
4. The key techinical contribution of the paper is the development of the hallucination attack, i.e. use a gradient-based token replacing strategy from [3] to recursively obtain the propmt candidate set with (weak semantic) / without (OoD) the metric constraints with the original sample. The token-replaced strategy seems to be a widely-used strategy already, and the author's work seems to change the objective from classical tasks (e.g. sentiment analsyis) into hallucinated response. I would like the author to better explain their techinical novelty compared to the existing works.

**Questions:**

1. How is the entropy threshold defense empirically used? Rejecting 46.1% OoD prompts and 61.5% weak semantic prompts seems to be a very low ratio. I think if the author train a supervised classifier on response prompts with three labels raw, OoD, weak semantic prompts can probably give a higher performance. I don't think thresholds is a realistic defense strategy.
- If we train the adversarial prompts differently, e.g. let the norm constraint be the entropy constraint, this defense strategy will likely fail very bad.
2. I am wonderly if it is possible to try other gradient-based attacks than the token replacing attacks?

[1]: Siren's Song in the AI Ocean: A Survey on Hallucination in Large Language Models. Zhang et al.

[2]: Adversarial Examples Are Not Bugs, They Are Features. Ilyas et al.

[3]: Universal adversarial triggers for attacking and analyzing nlp. Wallace et al.

I am happy to raise my rating if the authors fully address my concerns.

---

### Official Review · Reviewer_Vsro · 2023-11-01

**Soundness:** 2 fair
**Presentation:** 1 poor
**Contribution:** 2 fair
**Rating:** 3
**Confidence:** 2

**Summary:**

This paper introduces a hallucination-triggering approach in LLMs based on weak-semantic and out-of-distribution (OOD) prompting. The paper formalizes adversarial attacks by perturbing the prompt using a gradient-based token-replacing strategy. The paper evaluates the technique using a hallucination-triggering success rate and presents qualitative examples of weak-semantic and OOD prompting results on Vicuna-7B and LLaMA2. The paper also presented an entropy threshold-based defense mechanism to the proposed adversarial attack strategy.

**Strengths:**

The paper presents a unique perspective to the problem of LLM hallucination and shares some compelling quantitative and qualitative results of targetted adversarial attack showcasing the attacked LLM to respond with a specific/predefined hallucination.

**Weaknesses:**

- There is no mention of the size of the dataset.
- There isn't any discussion on the effect of the proposed adversarial attack as a function of model size.
- In the evaluation, it isn't clear if the attacked LLM response is matched word by word to the predefined hallucination set or only certain "fabricated"/"non-existent" phrases are matched. If the former, the experimental setup would have benefitted from , as LLMs can be non-deterministic in their response.

- One of the major weaknesses I found with the paper was its structure and grammar. The paper can benefit from an improved organization of its content. For example: it could begin by broadly describing the overall setup and then going into the details. Reading Section 2 raised initial questions about experimental setup but some were explained later on. Related work towards the end makes the motivation of the work weak.

- There are many grammatical mistakes. Noting a few below:
  - does not consist *with* human cognition and facts
  - Moreover, this also *formulate* a new paradigm in the community
  - But there is few work *contribute* to the direction

**Questions:**

- In section 3.1, what does 'fit' mean in "we fit it into the LLMs and respond with a correct answer f(x) ∈ T , i.e., “Joe Biden was the victor of the United States presidential election in the year 2020”?

- What's the size of D and ˜D?

- How were the hyperparameters setting notes in Section 4. selected?

- Were there ever more than 1 weak-sematic attacked prompts ˜x returning the same ˜y?

**Details Of Ethics Concerns:**

-

---

### Official Review · Reviewer_7xzS · 2023-11-05

**Soundness:** 2 fair
**Presentation:** 2 fair
**Contribution:** 2 fair
**Rating:** 5
**Confidence:** 3

**Summary:**

This paper proposes the use of adversarial attacks to elicit factually incorrect outputs from LLMs and presents it as an automatic method for triggering hallucinations. To this end, the authors have constructed a dataset of question-and-answer pairs with fabricated answers and employed discrete adversarial attacks to trigger the model into generating fabricated outputs. The adversarial attacks in this paper have two variants: one aims to minimally alter the original question to maintain semantic integrity, and the other modifies randomly initialized prompts without semantic constraints. The authors tested their methods on Vicuna and LLaMA2 and demonstrated that they could trigger the desired outputs with high probability. They also discuss a defense method using perplexity.

**Strengths:**

1. This paper demonstrates that adversarial attacks can be employed to elicit desired fabricated responses from LLMs, presenting an intriguing application of such attacks.
2. The visualizations and examples in this paper help readers understand the concept.

**Weaknesses:**

1. My primary concern is that the adversarial attack methods used in this paper appear to be very similar to those of Shin et al. (2020) and Zou et al. (2023). The optimization process, including gradient backpropagation and random position selection, seems to be identical to that of Zou et al. The only difference I discern is that the authors have added a perturbation budget constraint in their "weak semantic" setting in an attempt to maintain semantic consistency (the exact implementation of this hard constraint in the optimization process is unclear). Despite these similarities, the authors neither cite nor discuss these related works, nor do they compare their results with them in the experiments. I suggest that the authors thoroughly discuss the similarities and differences with these related papers and consider using them as baselines for their experiments.
2. While "hallucination" may be an overused term today, arguably, people are more concerned with why models produce factually incorrect or inconsistent answers under "normal," "meaningful" prompts. Even for weak semantic prompts, the prompts obtained in this paper still seem to have a clear discrepancy from normal human requests. Given this, the results of the paper are not particularly interesting, as Zou et al.'s adversarial attacks have already demonstrated that models can output strings that perfectly match their expected outputs. I recommend that the authors discuss more how their work helps understand the phenomenon of hallucination and better connect their results with the mechanisms of hallucination.
3. The experimental results are not comprehensive enough. The authors did not consider baseline methods (e.g., from Shin et al. or Zou et al.), making the results difficult to interpret. The experiments were conducted on two open-source models, whereas people might be more interested in the performance on proprietary models such as ChatGPT and Claude 2.
4. The evaluation metrics in the experiments are also unclear: the authors mention using "human feedback to assess whether LLMs' responses are satisfactory," but what about partial matches? Moreover, they use humans as semantic extractors, but how do they evaluate whether the features extracted by humans meet the ε budget? The authors could clarify some key details of the evaluation metrics.
5. The presentation of this paper could be significantly improved. Some sentences are difficult to understand, such as the sentence on page 4: "For an original prompt x, the key idea is to selectively replace some “trigger” tokens τ with several iterations," and "a sentence x is mapping from some sequence of tokens." Additionally, there are numerous spelling errors, such as on page 2, "we also conduct heuristics experiments on defensing hazard hallucination attack," heuristics -> heuristic, defensing -> defending against. I suggest the authors proofread the paper carefully to make it more readable for the audience.


[1] Shin et al., AutoPrompt: Eliciting Knowledge from Language Models with Automatically Generated Prompts, EMNLP 2020

[2] Zou et al. Universal and Transferable Adversarial Attacks on Aligned Language Models, arXiv 2307.15043

**Questions:**

Please refer to the weakness section.

**Details Of Ethics Concerns:**

~Some examples in the paper may be offensive to some readers.~

Clarification: Examples such as "the United States declared war on the Islamic Caliphate" in "2022" might be sensitive under the current situation and may warrant a warning at the beginning of the paper.